# Edible Insects: A Historical and Cultural Perspective on Entomophagy with a Focus on Western Societies

**DOI:** 10.3390/insects14080690

**Published:** 2023-08-04

**Authors:** Marianna Olivadese, Maria Luisa Dindo

**Affiliations:** Department of Agricultural and Food Sciences, University of Bologna, Viale Fanin, 42, 40127 Bologna, Italy; marianna.olivadese2@unibo.it

**Keywords:** entomophagy, historical perspective, cultural diversity, novel food, sustainability

## Abstract

**Simple Summary:**

Entomophagy has a long and rich history in human culture. In fact, insects have been a part of human diets for thousands of years, with evidence of their consumption found in prehistoric archaeological sites. Throughout history, entomophagy has been a common practice in many cultures, particularly in parts of Africa, Asia, Latin America, and Oceania. In some societies, insects were considered a delicacy and were reserved for special occasions, while, in others, they were a staple food source. The roots of entomophagy vary depending on culture and region, but common reasons include the nutritional benefits of insects, their abundance and accessibility, and the cultural and religious significance of certain species. While the practice of entomophagy has declined in some parts of the world due to the influence of Western culture and industrialization, it continues to be important in many societies. Despite its long history and potential benefits, entomophagy has faced cultural and social stigmas in many parts of the world. However, recent efforts have been made to promote entomophagy as a sustainable and nutritious food source and to challenge cultural biases against insect consumption.

**Abstract:**

The relationship between insects and humans throughout history has always been complex and multifaceted. Insects are both a source of fascination and fear for humans and have played important roles in human culture, economy, and health. Nowadays, there is growing interest in using insects as a sustainable and environmentally friendly source of protein and other nutrients. Entomophagy can be seen as a new opportunity for the food industry and global food security. In fact, insects require far fewer resources than traditional livestock, and there are many references to insect consumption in human history. The ancient Romans are known to have eaten various insects, including beetles, caterpillars, and locusts. Insects such as crickets, grasshoppers, and ants have been eaten for centuries and are still considered a delicacy in many parts of the world, especially in Africa, Asia, Latin America, and Oceania. Entomophagy has, thus, been a part of human history for thousands of years and continues to be an important food habit for many people around the world. These topics are explored in this article from a historical and cultural perspective (e.g., ecological, nutritional, spiritual, and socio-psychological), with a focus on the progressive acceptance of edible insects in Western societies, since this novel food has also its roots in the Western world.

## 1. Introduction

The relationship between humans and animals has varied throughout history and across different cultures. For centuries, humans have hunted animals for food or fur, besides using them for transportation or labor. In addition, animals have been caged for people’s pleasure or tamed to be exploited in various ways. In particular, the domestication of animals such as cows, sheep, and horses enabled humans to establish settled agricultural societies, thus playing a significant role in human history by providing benefits in terms of food, transportation, farming, and companionship [1].

Throughout history, animals, including insects, have often been imbued with spiritual significance and regarded as sacred or symbolic beings. Even nowadays, they are often seen as symbols of the natural world; as such, they represent our connection to the Earth and the cycles of life and death. They can also symbolize the seasons, the elements, and the different forces of nature and have, therefore, been associated with specific qualities and traits, such as strength, wisdom, speed, or cunning [2]. For example, in ancient Egypt, the scarab beetle was associated with the god Khepri and represented rebirth and renewal [3]. In Hinduism, cows are considered sacred and should not be harmed [4].

Specifically, Brahmins never eat or handle any meat, fish, or eggs, while people belonging to the other three castes (Kshatriya, Vaisya, and Shudra) sometimes eat fish, eggs, and even chicken, goat, or mutton, but never during religious occasions [5].

In general, the relationship between humans and animals has been complex and even contradictory. On the one hand, humans have always relied on animals for survival, but, at the same time, they have also harmed them through overhunting, pollution, and habitat destruction. Today, however, there is growing recognition of the need to strike a balance between human needs and preserving the natural world. There are many things we can do to preserve the balance crucial for the long-term sustainability of our planet [6]. For example, we can make conscious decisions to reduce our carbon footprint by adopting “green” habits such as using renewable energy, reducing waste, conserving water, and using public transportation or walking instead of driving; we can also promote biodiversity by supporting the planting of native plants and the conservation of natural habitats. In addition, we can reduce the consumption of resources such as energy, food, and water [7].

Through these actions, we can help ensure that human needs are met while preserving the natural world for future generations. In this framework, the choice of alternative protein and other nutrient sources, e.g., insects, should also be considered, as Meyer-Rochow had already suggested to the Food and Agriculture Organization (FAO) and World Health Organization (WHO) nearly 50 years ago [8]. It is well known that insects require significantly fewer resources than conventional livestock and may, therefore, be regarded as a sustainable food source [9]. Furthermore, entomophagy, i.e., the practice of eating insects by people [10], has been accepted in many societies worldwide throughout history, and, at the same time, insects are considered an emerging food in areas where the use of eating insects is not a recent culinary tradition [11]. In this context, this review approaches the topic of entomophagy from a historical and cultural perspective, as this novel food is rooted in the past, but, at the same time, it is projected into the future and may become increasingly important in the face of global population growth and changing climate conditions [12,13].

## 2. Insects in the Human Diet throughout History

Insects were probably an important food source for early human populations, and the evidence suggests that they were eaten as part of the human diet as early as prehistoric times, although ancient insect consumption is not easy to prove due to poor preservation [14,15]. The remains of insect integuments, wings, and other body parts have, however, been found in the fossilized feces (coprolites) of ancient humans in caves in the USA and Mexico [16]. Cave art, dating back tens of thousands of years, provides valuable insights into the beliefs, practices, and daily life of prehistoric human societies. Although large, hunted animals such as bison, mammoths, and deer are commonly depicted in cave paintings, insects also appear in some of these works [17]. Bees, wasps, beetles, and other flying insects are often depicted in hunting scenes, feeding on plants or entangled in webs. The symbolism behind these insect depictions is not always clear, but they probably had meaning for the artists and their communities [18]. Incidentally, the consumption of insects likely played a crucial role in the diet of early humans due to their abundance and nutritional value [19].

In south-western France, in the Magdalenian cave of Les Trois Frères (Upper Paleolithic, dating from around 17,000–12,000 years ago [20]), there is a remarkable representation of a ‘cave grasshopper’ engraved on an animal bone. The presence of this engraving has been interpreted by some researchers as a possible link between insects and feeding practices [21]. Another archaeological site in Europe that provides insights into insect consumption in prehistoric times is Altamira, a famous cave in northern Spain known for its remarkable paintings of animals, such as bison and horses, as well as abstract designs. The paintings provide insights into the daily life and beliefs of the prehistoric peoples who inhabited the region during the Upper Paleolithic, some 36,000 years ago, and also depict a collection of edible insects and nests of wild bees [22].

It is important to recognize that the study of entomophagy by humans in Europe is an evolving field, and ongoing research and new discoveries may shed more light on prehistoric insect consumption. Traditional insect consumption practices may have been more common than archaeological evidence currently suggests [23].

More recently, many cultures around the world, including the Ancient Near East, have been consuming insects as a regular part of their diet [24]. There are several references to the consumption of insects in the Bible, particularly in the Old Testament. In *Leviticus* 11:20–23, for example, it is written that any winged insect that walks on “four” legs is considered unclean, while other winged insects, such as locusts, crickets, and grasshoppers are considered clean and can be eaten by the Israelites [5,25,26,27]: 

“All flying insects that walk on all fours are to be regarded as unclean by you. There are, however, some winged creatures that walk on all fours that you may eat: those that have jointed legs for hopping on the ground. Of these you may eat any kind of locust, katydid, cricket, or grasshopper. But all other winged creatures that have four legs you are to regard as unclean”.

Since insects have six legs, it is not entirely clear why insects with “four” legs were singled out as unclean while other winged insects were considered clean. One possibility is that this classification was based on observable characteristics of insects that were known to the ancient Israelites. Insects such as beetles (probably considered unclean) have a different body structure and mode of movement compared to other insects, such as grasshoppers or locusts, which have powerful hind legs and can jump. Another possibility is that the classification of insects with “four” legs as unclean was based on cultural or religious beliefs. In many ancient cultures, insects were associated with death, decay, and impurity; therefore, they were considered unclean. This may have been the case for the ancient Israelites as well. In the end, the reason for the inclusion of these particular insects in the list of unclean animals is not entirely clear, as the text does not provide a specific explanation. However, some scholars suggest that it may have been due to their association with arid and dusty environments, which were considered unclean in ancient Israelite culture [28].

In addition, in the New Testament, John the Baptist is described as having eaten locusts and wild honey in the wilderness (Matthew 3:4): “*Ipse autem Joannes habebat vestimentum de pilis cameli, et zonam pelliciam circa lumbos ejus: esca autem ejus erat locustae et mel silvestre*” (=“John’s clothes were made of camel’s hair, and he had a leather belt around his waist. His food was locusts and wild honey”) [29].

In ancient Greece, the consumption of insects was not widespread or prominent compared to other food sources. The ancient Greeks primarily relied on a diet that consisted of grains, fruits, vegetables, legumes, fish, meat (mainly lamb and pork), and dairy products. Insects were not a significant part of their culinary practices or cultural traditions [30]. However, it is worth noting that some historical sources mention the use of certain insects as food and feed in ancient Greece, albeit to a limited extent. For example, the philosopher and naturalist Aristotle wrote about using *Cossus cossus* (L.) caterpillars as bait for fishing. He noted that the caterpillars were especially effective at attracting fish and were commonly used by fishermen in his time. In Book V of “The History of Animals” [31], Aristotle described the life cycle and habits of the *Cossus* caterpillar, which, according to him, was commonly found in the Mediterranean region. He mentioned that *Cossus* caterpillars were used by the ancient Greeks as a source of food, particularly by those living in rural areas. Aristotle also noted that *Cossus* caterpillars were considered a delicacy by some people, who would roast or boil them before eating them [32]. He described their flavor as sweet and nutty, with a mushroom-like texture [33].

Insects were normally consumed in ancient Roman society and were considered a delicacy, on one side, and food for poor people, on the other side [34]. Pliny the Elder, a Roman author and naturalist who lived in the first century AD, wrote extensively about the natural world and the use of plants and animals for food and medicine. In his writings, he mentioned the consumption of insects by the ancient Romans, particularly during times of food scarcity [35]. Pliny the Elder, in his book “*Naturalis Historia*”, noted that locusts and grasshoppers were considered a delicacy and were often prepared by roasting or frying. He also mentioned the consumption of beetles, ants, and caterpillars, and described the use of ants in medicine for their supposed healing properties [36].

Pliny wrote about the Roman practice of eating beetle larvae, which were considered a delicacy and were fattened up in special jars. He also described the consumption of “*Cossus*”, possibly the larvae of *C. cossus*, which were considered a delicacy by the Roman elite. Pliny noted that *Cossus* was consumed with great relish and was a very tasty food. He noted that these insects were a food source for the poor but were also consumed by wealthy people.

While Pliny recognized the nutritional value of insects and their potential as a food source, he also noted that some insects (not described) could be harmful if consumed and warned against eating certain species. Nevertheless, his writings demonstrate that the consumption of insects was not uncommon in ancient Roman society.

Information about *Cossus* and other edible insects is given in Pliny (Book 17, Chapter 37). 

“Particular trees are attacked by worm in a greater or smaller degree, but nearly all are liable, and birds detect worm-eaten wood by the hollow sound when they tap the bark. Nowadays indeed even this has begun to be classed as a luxury, and especially large wood-maggots found in oak-wood—the name for these is *cosses*—figure in the menu as a special delicacy, and even these creatures are fed with flour to fatten them for the table. The trees most liable to be worm-eaten are pears, apples, and figs; those that have a bitter taste and a scent are less liable. Of the larvae found in fig-trees some breed in the trees themselves, but others are produced by the insect called in Greek the horned insect; all of them, however, assume the shape of that insect, and emit a little buzzing sound. Also, the service-tree is infected with red, hairy caterpillars, which eventually kill it; and the medlar as well is liable to the same disease when it grows old” [37].

Then, Pliny (in Book 29, Chapter 16) describes how the taste for luxury led to the consumption of various exotic foods, including large worms found in the roots of trees. The Latin text of the passage is: “*Iam quidem et hoc in luxuria esse coepit, praegrandesque roborum vermes, ut iam non solo silvarum latebras, sed etiam urbes petantur, et pretiosus quidquid non gustaveris*” [=“Now indeed even this has begun to be regarded as luxury, and huge worms found in the roots of trees are sought not only in the depths of forests but even in the cities, and anything you have not tasted becomes a delicacy”] [38].

The previously mentioned “horned” insects were probably beetles belonging to the genus *Lucanus* [39]. Pliny often mentioned this genus, which includes the stag beetle, in his *Naturalis Historia*. In Book 11, Chapter 52, he describes the stag beetle as a large and fearsome insect with powerful mandibles that it uses to crush fruits and flowers. He noted that the beetle is commonly found in wooded areas and suggested that it had a short lifespan, living for only a few weeks in its adult form. Pliny’s description of the stag beetle is remarkable for its accuracy, given the limited scientific knowledge of the time. The name *Lucanus* comes from the Latin word “*lucere*” meaning “to shine” [40] and probably refers to the shiny appearance of the beetle’s body. The genus *Lucanus* is still recognized today and includes over 1200 species of stag beetles found globally throughout the world [41].

Pliny, however, mentioned several other types of “horned” insects in his *Naturalis Historia*. For example, in Book 11, Chapter 48, he described the horned beetle (*Scarabaeus*) and its habits. In Book 21, Chapter 27, he mentioned the horned fly (*Tabanus*), which he described as having a sharp sting and being particularly troublesome to livestock. In Book 29, Chapter 11, he described the horned drone (*bombilator*), which (he said) had two curved horns and made a buzzing sound like a trumpet.

Other mentions of “horned” insects included the short-horned grasshopper (*Locusta*) (referring to antennae), the horned cicada [(*Centrotus cornutus* (L.)], which displays two lateral horns on the thorax and a long backward-directed spine, and the horned chafer (probably the stag beetle, the male of which displays two horned mandibles). Pliny was a keen observer of nature, and his *Naturalis Historia* is an important source of information on the animals, plants, and minerals known to the ancient world [36].

Pliny was not the only writer to discuss the consumption of insects: in fact, careful philological research suggests several references to the consumption of insects in Latin literature. Aelianus, a Roman author who lived in the second to third century AD and used to write in Greek, wrote extensively on the natural history of animals, including insects. In his work, “On the Characteristics of Animals” [42] (*De Natura Animalium*), Aelianus provided detailed descriptions of locusts and the ways humans consumed them. According to this author, locusts were considered a delicacy in some parts of the ancient world, particularly in the Middle East. He described how locusts were caught in large nets and then either roasted or boiled before being consumed. He also noted that the wings and legs were removed before cooking and that the head and body were eaten separately. Aelianus also described the taste of locusts, which—he said—was “a little like that of a fish, but more pleasant”. He also noted that locusts were believed to have medicinal properties and were used to treat various ailments, including headaches, insomnia, and coughs [43].

It is worth noting that the Romans did not necessarily eat insects as a staple part of their diet, but also as a delicacy or novelty item. Insects were often consumed at banquets and other special occasions, and their consumption was associated with luxury and sophistication [44]. The poet Ovid (Publius Ovidius Naso, 43 B.C.–17/18 A.D.) referred to the consumption of *cossus* caterpillars in his work. In his poem “The Art of Love”, he wrote about the culinary delicacies enjoyed by the wealthy elite, including “caterpillars of the cossus moth, served on silver dishes” [45]. Additionally [*Tristia* (Book 3, Elegy 10, vv. 21–22)], he described the practice of consuming cicadas, as well as crickets. There are several other references to the consumption of insects throughout Ovid’s *Metamorphoses*, demonstrating the prevalence of this practice in ancient Rome (Book II, Line 734). In other passages (*Metamorphoses*, Book VIII, Lines 620–623 and 736–747), Ovid described a famine in which people were forced to eat insects (such as beetles and grasshoppers) to survive, thus proving that insects were not only a delicacy but also a source of food during times of scarcity and hardship. This “emergency” food could, however, provide enough energy “to hunt and carry heavy loads” [*Metamorphoses*, (Book XIV, Lines 701–705)] [46].

The poet Horace (Quintus Horatius Flaccus 65 B.C.–8 B.C.) also wrote about eating beetles, praising the simple pleasures of life, such as consuming insects and drinking wine, over the excesses of the rich [47]. Likewise, Lucretius (Titus Lucretius Carus, 99–55 B.C.) (*De Rerum Natura*, Book 5, Lines 1423–1428) described the consumption of cicadas, which were apparently eaten in ancient Rome in a similar fashion as grazing cattle [48]. The Roman poet Martial (Marcus Valerius Martialis 40–102/104 A.D.) (Book 13) mentioned locusts in several of his epigrams. One of his most famous locust references appears in Epigram 84 from Book 14. In this epigram, Martial used locusts to symbolize abundance and prosperity, contrasting them with the meager subsistence of rural peasants. Martial wrote a few verses about eating locusts, although his writings did not specifically discuss the nutritional value of locusts. In Epigrams Book 13, Martial described how a wealthy friend served locusts as a delicacy at a banquet. The tone of the poem is light and humorous, and Martial seems to be poking fun at his friend’s extravagant tastes [49]. Moreover, Martial mentioned the culinary uses of locusts in Epigram 95 from Book 13. In this epigram, Martial praised a man who could enjoy locusts as tasty and nourishing food. The description of the locust-eater’s other culinary preferences, such as tortoise eggs and small insects, suggests that he was not a wealthy man, but he could enjoy the simple pleasures of life [50].

It is also possible that military personnel in ancient Rome ate insects, particularly in times of famine or when on campaign in regions where traditional food sources were scarce or unavailable. For example, Pliny [51] wrote about the consumption of locusts by the Roman army during the Third Punic War (149–146 B.C.), when food supplies were scarce. He noted that the soldiers boiled the locusts and then pounded them into a paste, which they mixed with grain to make bread. Additionally, there are records of Roman soldiers consuming beetles, ants, and other insects during military campaigns in Africa and other parts of the empire [52,53]. Therefore, while insects were likely not a staple of the Roman military diet, they may have been consumed out of necessity in certain situations.

More evidence of the fact that the consumption of insects was a common practice in Roman society is found in *Satyricon*, a novel by the Roman writer Gaius Petronius Arbiter (27 A.D.–66 A.D.), which includes a famous scene describing the consumption of insects at a banquet hosted by the wealthy and ostentatious Trimalchio [54]. In this scene, Trimalchio, an extravagant freedman, serves a dish of roasted beetles and grasshoppers, which his guests consume with relish. The scene is a satire of the excesses of Roman society, and the consumption of insects is just one example of the lavish and extravagant foods served at Trimalchio’s banquet. Trimalchio encourages his guests to try the dish, even suggesting that it is a delicacy (*Satyricon*, 31.8), and, when some of the guests are disgusted by the idea of eating insects, he, as a comical and exaggerated character, insists that grasshopper and locust are also considered a delicacy among the Greeks.

In “*Satyricon*” Petronius portrayed the decadence and excesses of the Roman elite, particularly when it came to their dining habits. The characters in the novel consume a wide variety of exotic and unusual foods, often to the point of excess and even nausea [55]. This suggests that Petronius might have viewed the consumption of insects as just another example of the extravagant and frivolous behaviors of wealthy and powerful people. Furthermore, Petronius was known for his satirical and subversive writing, which often poked fun at the social norms and conventions of ancient Roman society. He may have used the idea of consuming insects to satirize or critique the culture of his time.

Despite the satirical nature of the scene, the consumption of insects was a common practice, both as food for the poor and as a curiosity for the wealthy. The scene also reflects the fact that Roman cuisine was highly diverse and included a wide range of foods, many of which were not commonly consumed in other parts of the ancient world.

## 3. The Consumption of Insects in the following Centuries in European Society

The study of insect consumption in ancient Rome is a fascinating area of research that offers insights into the dietary habits and cultural practices of this ancient civilization. With regard to the following historic periods, it is important to note that the consumption of insects during the Middle Ages was not well documented, and the available historical records are limited [56]. Therefore, our understanding of the extent and frequency of insect consumption during that period is somewhat limited. In summary, while there is evidence to suggest occasional insect consumption in certain contexts during the Middle Ages, entomophagy was not a prevalent or widespread practice throughout Europe. Insects were not a significant part of the everyday diet, and their consumption was often driven by specific circumstances or local customs [57].

After the medieval period, there were many references to the consumption of insects across a wide range of cultures and geographic regions, and there are several authors and works that discuss the consumption of insects [58]. In the European context, during the Renaissance period, the Italian physician and naturalist Andrea Bacci (1524–1600) wrote about the culinary uses of insects in his book “*De Thermis*”, published in 1571. Bacci recommended the consumption of locusts and cicadas as a healthy and nutritious food source [59,60].

The French physician and herbalist Pierre Belon (1517–1564) also wrote about the consumption of insects in his book “*Les Oeuvres de Pierre Belon*”, which was published in 1555. Belon described how the people of Turkey and other Eastern countries ate locusts, grasshoppers, and other insects, and recommended the consumption of ants as a remedy for indigestion [61]. Similarly, the Swiss physician Conrad Gesner (1516–1565) wrote about the medicinal and culinary uses of insects in his book “*Historiae Animalium*”, which was published in the mid-16th century. Gesner described how the people of Switzerland and Germany ate grasshoppers and locusts and recommended the consumption of beetles and ants as a remedy for various ailments [62].

Overall, while the consumption of insects during the Renaissance period was not as widespread as it was in earlier times, there were still several authors and scholars who wrote about the culinary and medicinal uses of insects.

Ulisse Aldrovandi (1522–1605) was an Italian naturalist and physician who lived during the 16th century. He is known for his extensive studies of plants, animals, and minerals, which he documented in several books. In his book “*De Animalibus Insectis Libri Septem*” (Seven Books on Insects), Aldrovandi discussed the consumption of insects by humans. He observed that, in many cultures, insects were an important source of protein and other nutrients and were often used as food [63,64]. Aldrovandi recognized the potential nutritional benefits of consuming insects and even described the taste and culinary uses of certain species. He believed that insects could provide a cheap and plentiful source of food for the poor and that they could be farmed for this purpose. Aldrovandi talked about the consumption of insects in his book because he recognized their potential as a nutritious and sustainable food source and because he was interested in studying all aspects of the natural world, including the role of insects in human societies and ecosystems. In fact, Aldrovandi’s interest in insects was not just limited to their nutritional value. He also studied their anatomy, behavior, and ecology, and was one of the first scientists to recognize the importance of insects in pollination and other ecological processes [64].

Subsequently, the famous biologist Charles Darwin (1809–1882) wrote about his experiences eating insects during his travels in South America in his book “The Voyage of the Beagle” [65]. Moreover, the French naturalist Jean Henri Fabre (1823–1915) wrote about the culinary uses of insects in his book “Souvenirs Entomologiques” [66].

Vincent M. Holt, an English naturalist and author who lived in the 19th century, wrote a book in 1885 entitled “*Why Not Eat Insects?*” in which he argued that insects were a nutritious and sustainable food source [67]. Holt compared insects to other widespread foods, such as eels (whom he called “the scavenger of the sea”), octopus, cuttlefish, and oysters, wondering what the basis of the disgust caused by insects was. He also argued that one way to succeed in popularizing insect consumption might be to make them a fashionable food, fashion being the most powerful motivator in this world.

More recently, Paul Rozin, an American psychologist born in 1936, wrote extensively about food and culture. In several of his works, he discussed the history and cultural significance of insect consumption [68]. He compared attitudes towards entomophagy in various cultures to identify commonalities and differences in their practices and beliefs in order to provide a deeper understanding of food preferences and human behavior in different societies [69]. His research may have had implications for nutritional science, biology, and entomology, shedding light on a lesser known but potentially valuable food source.

In summary, European scholars from different eras discussed insect consumption by providing historical context, presenting a cross-cultural perspective, and conveying the cultural significance of this practice. They contributed to bridging the gap between the past and present, fostering a deeper understanding and appreciation of the cultural perspective on insect consumption.

## 4. Examples of Insect Consumption in Non-Western Countries Today

Cross-cultural considerations are important when discussing the use of insects as human food. Each cultural context brings its own traditions, beliefs, and culinary practices. Although this review focuses primarily on Western societies, it is important to approach these topics with respect for cultural diversity and to recognize the importance of local knowledge and traditions [70]. Nowadays, more than 2000 insect species are consumed around the world [71], and some examples concerning non-Western countries are given here. In African literature, the consumption of caterpillars belonging to the Cossidae family (“the goat moths” or “carpenter moths”) has been a common theme in traditional stories and folklore. In some cultures, the caterpillars are believed to have special powers or spiritual significance and are used in rituals and ceremonies [72]. These large caterpillars feed on various trees, including oak, willow, and apple trees. In some African countries, they are collected from wild trees and sold in local markets as a food source. The caterpillars are typically roasted or fried and eaten as a snack or as part of a meal. They are also believed to have medicinal properties and are sometimes used to treat a variety of ailments [9,73]. There is a significant amount of literature on the consumption of mopane worms, which has focused on their nutritional value, cultural significance, and potential as a sustainable food source [74]. Mopane worms are the larvae of the emperor moth *Gonimbrasia belina* (Westwood), native to sub-Saharan Africa, and are commonly eaten in southern Africa, particularly in Botswana, Zimbabwe, Zambia, and South Africa. Mopane worms are a rich source of protein and other nutrients and are an important food source for many people in the region. They are traditionally harvested from mopane trees, which are common in the region, and are often eaten either boiled or fried [75].

In South America, the larvae of the palm weevil *Rhynchophors palmarum* L., known as “suri”, are a traditional food source for indigenous communities in the Amazon rainforest. They are often eaten raw or roasted and are sometimes used in traditional medicines as well [76].

In Southeast Asia, so-called “bamboo worms” are a popular street food in Thailand, Laos, and Vietnam. *Omphisa fuscidentalis* (Hampson) is the scientific name of the species, including a moth belonging to the family Crambidae, which includes more than 11,000 described species worldwide [77,78]. In Northeast India, insects such as silkworm pupae, bamboo worms, and red ants are commonly consumed in certain tribal communities, and they are often cooked with spices, creating unique flavors and textures in traditional dishes [79,80,81]. The Asian giant hornet (*Vespa mandarinia* Smith) has traditionally been reared and semi-domesticated by tribal people in Nagaland to be used as human food and a remedy for disease [82]. Oceania has a long history of incorporating insects into the human diet. Insects such as sago grubs and crickets have been commonly consumed and considered a valuable protein source in Australia, New Zealand, and Papua New Guinea [79,81,82]. Thousands of years ago, Aborigine people used to semi-cultivate cerambycid beetles by placing ovipositing females on suitable trees from which the larvae could later be harvested and employed as food. This “intentional” production of edible insects highlights a sophisticated understanding of the interconnections between plants, insects, and food sources for humans [81]. In Papua New Guinea, insects have been consumed as a traditional food source for centuries. They are often collected from the wild, and the selection of insect species and the methods of preparation and cooking vary among different ethnic groups [5,83,84]. The preservation and enhancement of these traditional practices are crucial for the conservation of cultural heritage and biodiversity and may contribute to a more holistic and sustainable approach to food systems and environmental management.

## 5. A Future Food from the Past in Western Societies

When comparing the consumption of insects in the past and today, different points of view offer insights into the changes and developments surrounding insect consumption over time. In the past, even in Western countries, insect consumption was rooted in cultural practices and traditions. As shown in previous paragraphs, entomophagy was considered a normal and accepted part of the diet in many contexts (while other insects were consumed out of necessity during times of food scarcity).

Today, while insect consumption is gaining traction globally, cultural attitudes toward eating insects still vary significantly. Some cultures embrace entomophagy as a traditional practice, while others may have reservations or consider it unconventional. With the growing recognition of the environmental impact of animal husbandry, there is an increased interest in insects as a more sustainable protein source. In fact, in terms of nutrition, insects are a rich source of protein, vitamins, and minerals and, with a better understanding of nutrition and advancements in food science, insects are being recognized for their nutritional value and potential health benefits [85].

Moreover, the present era has seen culinary innovations and technological advancements that have transformed how insects are consumed: they are now processed into various forms and incorporated into a wide range of food products, appealing to modern tastes and dietary preferences. This level of innovation and accessibility was likely not present to the same extent in the past. The current global perspective emphasizes the potential of insects to address food shortages and reduce the strain on resources [86]. The cultural and social attitudes towards insect consumption have evolved over time; the consumption of insects was likely more common and acceptable in biblical times and in ancient Rome than it is in modern European culture. They are generally not considered a mainstream food source, and insect consumption is relatively uncommon in many Western cultures, even if they are a sustainable food source for several reasons. Insects produce less waste and pollution and can be produced in a more efficient and environmentally friendly manner than traditional livestock. Insects can also be raised on organic waste, reducing the need for landfills and thus improving waste management [87]. Insects require significantly fewer resources than traditional livestock farming, resulting in a lower carbon footprint. They require less land and feed and produce fewer greenhouse gas emissions. They are highly efficient at converting feed into edible biomass. For example, crickets require six times less feed than cattle to produce the same amount of protein. They can be raised on vertical farms or in small spaces, reducing the need for large amounts of land. Therefore, they may even be produced in urban areas where space is limited, considering they require significantly less water than traditional livestock farming [88]. For example, crickets require 2000 times less water than beef cattle to produce the same amount of protein. Insects are less susceptible to disease and do not require antibiotics or growth hormones. This reduces the use of chemicals in their production, which can harm the environment and human health. The consumption of insects can thus offer a range of benefits for the future, including improved nutrition, environmental sustainability, food security, cultural diversity, and innovation in the food industry.

Insects may be considered comparable in nutritional value to conventional food [89] and may also be digested by most human beings [90]. They are a rich source of protein, vitamins, and minerals. They contain all the essential amino acids required by the human body, making them a high-quality protein source. Insects are also low in fat and have a favorable ratio of omega-3 to omega-6 fatty acids [9,91].

As the world population grows, there is an increasing need for sustainable food sources. Insects can provide a reliable and sustainable source of protein. As shown previously, in many cultures around the world, insects are already a traditional food source. Incorporating insects into global food systems, including in Western societies, may help preserve traditional food cultures and promote cultural diversity [92]. In recent years, there has been a renewed interest in the consumption of insects and in novel food products since they are being developed with sustainability in mind, using ingredients that are locally sourced, organic, and/or regeneratively produced; these production methods help to minimize environmental impacts and support sustainable agriculture in the context of a circular economy. Insect production, however, requires expertise, since different factors, including the quality of the diet offered and microbiological assessment, need to be taken into account [93,94,95].

Novel food refers to any food or food ingredient that was not consumed to a significant degree within the European Union (EU) prior to 15 May 1997. This includes either newly developed foods/food ingredients or traditional foods from other parts of the world that are not currently consumed within the EU. The importance of novel food lies in its potential to offer new and innovative food products to consumers while ensuring that they are safe and do not harm human health. Novel food can provide a range of benefits, such as improved nutrition, enhanced flavor, and new culinary experiences. However, the safety of novel food is paramount; therefore, it is subject to a rigorous safety assessment before it can be authorized for sale within the EU. The safety assessment evaluates the potential risks associated with the food, including its composition, toxicity, and any possible allergenic effects [96]. The EU’s regulation of novel foods provides a framework for the authorization, safety assessment, labeling, and traceability of novel food products. This ensures that consumers can make informed choices about the food they eat and can have confidence in the safety and quality of novel food products [97,98].

In summary, the development of novel foods is an opportunity to explore more sustainable food sources and production methods. By promoting the use of sustainable and innovative food sources, novel foods have the potential to contribute to a more sustainable and resilient food system because they can help address the problem of malnutrition and improve the overall health of populations. A nutritious diet is essential for good health, and novel food can provide the essential nutrients needed for optimal health [99].

The production of novel food (including insects) may also create jobs and stimulate economic growth, particularly in rural areas where agriculture is an important productive activity. This can help to reduce poverty and promote economic development as well as promote a more ethical and sustainable food system [100].

In Europe, the consumption of insects as a food source has progressively declined with the spread of the cultural taboo against consuming “creepy crawlies”. The European Union has not yet fully approved entomophagy for widespread consumption, but there has been increasing interest and research in this area in recent years. Some EU member states (including Belgium and the Netherlands) have allowed the sale of certain insect products, but there is not yet a harmonized framework for regulating insect-based foods across the EU [101]. However, there are still some traditional dishes that include insects in European cuisine, such as the Sardinian delicacy of *Casu Marzu*, a cheese colonized by insect larvae (see below); moreover, in 2018, the European Food Safety Authority (EFSA) approved the use of whole and ground crickets as a novel food, meaning it can be used as an ingredient in various food products sold in the EU. Cricket flour is made from ground crickets and is a high-protein, gluten-free alternative to traditional flour. It can be used in various food products, such as protein bars, pasta, and baked goods. The approval of cricket flour as a food ingredient by the EU is a significant milestone for the insect food industry, as it opens up new market opportunities and increases the acceptability of insect-based foods among consumers. EU approval also means that cricket flour is subject to the same safety and labeling regulations as other food products sold in the EU, ensuring it is safe for human consumption [102].

*Casu Marzu* is a traditional cheese made in the region of Sardinia, Italy, that is made from sheep’s milk. What makes this cheese unique is that it is intentionally infested with the live insect larvae of the cheese fly, *Piophila casei* (L.), which are allowed to hatch and mature in the cheese. The larvae feed on the cheese and break down its fats, producing a soft and creamy texture. *Casu Marzu* has a pungent odor and is considered a delicacy by some people [103]. It has a long history in Sardinian culture and is believed to have originated as a way for shepherds to preserve cheese during the long winters when fresh milk was scarce. The practice of intentionally infesting cheese with insect larvae is also found in other Italian regions (such as Abruzzo and Apulia), as well as in Corsica and in other cultures, such as in the Philippines. Despite its popularity, the consumption of *Casu Marzu* is controversial due to health concerns. In fact, the larvae in the cheese can survive in the human digestive system and have been known to cause gastrointestinal problems, including vomiting and diarrhea. As a result, *Casu Marzu* is banned in many countries [104,105].

*Marcetto* cheese from Abruzzo is also known for its feature of being ripened with live cheese flies. The production of this typical food involves inoculating it with *P. casei* maggots, which are then allowed to feed on the cheese for several months. During this time, the cheese develops a pungent and earthy flavor that is highly prized by cheese connoisseurs. Another example is *Queso del Infierno*, a Spanish cheese that is made with goat’s milk and is ripened with cheese flies, following the same methods described above. The resulting cheese has a soft and creamy texture and a pungent flavor [106]. There are also other types of cheese that are made with insects or insect-derived ingredients, such as the use of bee pollen in some types of goat cheese, or the use of larvae or pupae in other types of cheese in Asia and Africa. These cheeses are often considered delicacies and are valued for their unique flavors and textures [107].

In recent years, cricket flour has become increasingly popular due to several factors, including its nutritional value, sustainability, and versatility in culinary uses. Regarding nutritional value, cricket flour is a good source of protein, containing up to 70% protein by weight, as well as vitamins and minerals such as iron and vitamin B12 [108]. It is also low in fat and carbohydrates, making it a popular choice among health-conscious consumers. Regarding sustainability [109], compared to traditional livestock such as cows or pigs, crickets require less space, water, and food to produce the same amount of protein. In addition, crickets emit fewer greenhouse gases and generate less waste than traditional livestock [110]. Regarding versatility, cricket flour can be used in a variety of food products, including baked goods, snacks, and protein bars. According to some consumers, it has a neutral flavor and can easily be incorporated into recipes without affecting taste or texture [111]. Moreover, the “novelty factor” may also have an impact: eating insects or their derivatives (including cricket flour) is considered unusual in many parts of the world, and some consumers are drawn to the novelty of trying a new and exotic food source. Since it is gluten-free, for people with specific intolerances, cricket flour is a viable alternative to wheat flour [112]. Overall, the nutritional, environmental, and economic benefits of cricket flour contribute to its growing popularity in the food industry.

## 6. Conclusions

By exploring the rich history of insect consumption, we can gain a greater appreciation of the important role that insects have played and, probably, will continue to play in human culture and nutrition.

Evidence indicates that humans have been consuming insects as a food source for tens of thousands of years. Exploring archaeological and anthropological sources, as well as potential reasons why early humans turned to insects as a food source through a cultural-historical perspective, we noted the nutritional benefits of insects, which were apparently valued by our ancestors, whether as an emergency or luxury food. In Western society, while the cultural acceptance and availability of insects may have been more widespread in ancient Rome, and, to some extent, in the Renaissance period, today’s interest in insect consumption is driven by sustainability, innovation, and the evolution of dietary practices. In ancient Rome, insects were consumed as a luxury food, but also as a response to food scarcity, partly due to their availability. They were a readily accessible source of nutrients and were collected during specific seasons or in times of need. Moreover, in ancient times, insect consumption was not governed by specific regulations or commercialized as a mainstream food industry. Today, in many Western societies, including those influenced by Roman culture, insect consumption is not so widespread or culturally accepted. However, interest in eating insects as a sustainable protein source is growing and gaining popularity in some communities and among adventurous eaters. Current interest in insect consumption is driven by sustainability, innovation, and the evolution of dietary practices.

In conclusion, the consumption of insects has a long and varied history that dates to prehistoric times. Despite cultural taboos and societal biases against insects as a food source [5], many cultures worldwide have traditionally consumed insects for their nutritional and medicinal properties [113].

Today, with the growing awareness of the environmental and economic benefits of insect consumption, the practice is gaining wider acceptance and popularity across the globe. However, regulatory and cultural barriers still exist, and there is a need for further research into the nutritional content and safety of insect-based foods.

## 7. Future Directions

Insect consumption is becoming increasingly popular and is expected to continue to grow. Possible future directions for insect consumption are the following. (1) The expansion of insects as a mainstream food source: while insects are already consumed in many parts of the world, they are not yet a mainstream food source in many other regions. In the future, there may be a greater push to promote insects as a healthy and sustainable food source on a larger scale. (2) Increased use in processed foods: insect derivatives, i.e., cricket flour, are already being used in various food products such as protein bars, snacks, and baked goods. As consumers become more familiar with insects as a food source, we can expect to see more processed food products incorporating insect-based ingredients. (3) Greater emphasis on sustainability: insects are often promoted as a sustainable protein source due to their low environmental impact compared to traditional livestock. In the future, there may be a greater emphasis on exploiting the sustainability of insect farming practices, exploring factors such as feed sources, water usage, and waste management [114]. (4) Development of new insect-based products: a wide range of insect-based products is already available, but we can expect the development of new products as insects become accepted as a food source. These may include new insect-based meat alternatives, beverages, and functional foods. (5) Increased research into the nutritional benefits of insects: while insects are known to be a good source of protein, there is still much to be learned about their overall nutritional content and potential health benefits. In the future, more studies are expected to be performed to determine the nutritional value of insects and their potential role in promoting health and preventing disease [115].

In conclusion, in Western societies, insects are an emerging food that is rooted in the past and has gained popularity in recent years due to its nutritional value, sustainability, and versatility. As the global population continues to grow and the demand for protein increases, insects may become an increasingly important food source [8].

Today’s world presents great contradictions. For, on the one hand, a part of the population faces the overproduction of food, food waste, obesity, and other health problems related to overeating, but, on the other hand, many people are still facing food insecurity, even in wealthy countries [116]. Insects can certainly represent a (not necessarily emergency) food for hungry populations, but they should, moreover, be regarded as a sustainable source of protein and other nutrients, the consumption of which can also help the wealthier part of the world to eat in a healthier and more sustainable way [9,74,117].

However, there are also challenges to be addressed, such as the cultural and regulatory barriers to insect consumption, as well as the need for more research into the nutritional content, production, and safety of insect-based foods [118]. With continued innovation and research, insects have the potential to play an important role in meeting the world’s food needs in a sustainable and nutritious way.

## Data Availability

Not applicable.

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
