# Peer review of "Edible Insects: A Historical and Cultural Perspective on Entomophagy with a Focus on Western Societies"

_insects, 2023, doi:10.3390/insects14080690_

Round 1

Reviewer 1 Report

Title: Edible insects: a historical and cultural perspective on entomophagy

Authors: Marianna Olivadese and Maria Luisa Dindo

This is a very well written, thorough and highly interesting and impressive manuscript. The authors cover historical facts that other reviews, with few exceptions, have in the past largely ignored.

I have rather few suggestions but ask the authors to consider them as acting on they would render their paper more balanced and complete. I cannot fully understand why there is a need for a ‘Simple Summary” and an “Abstract”. That is most unusual and I suggest to keep either the Summary or the Abstract.

On Line 10 the authors forgot to include Papua New Guinea, Australia, and New Zealand! In fact, West Australian Aborigines thousands of years already began to semi-cultivate edible grubs by deliberately placing egg-laying female cerambycid beetles on suitable trees from which people could later harvest the edible insects (Reim, H. 1962. Die Insektennahrung der australischen Ureinwohner. Akademie Verlag, Berlin  -  cited in: Meyer-Rochow, V.B. and Changkija, S. 1997. Uses of insects as human food in Papua New Guinea, Australia, and North-East India: cross-cultural considerations and cautious conclusions. Ecology of Food and Nutrition 36,159-185).

Line 55: the reference to ‘cows’ is a little odd, because to a Hindu, especially Brahmin, ALL animals and not just ‘cows’ are taboo and will not be eaten (I am married to an Indian lady!). There are reviews on Food Taboos in the Journal of Ethnobiology and Ethnomedicine, which could be of interest to the authors.

On line 70 after “…..should also be considered” continue the sentence and add “…as Meyer-Rochow nearly 50 years had already suggested to FAO and WHO”.  (Meyer-Rochow, V.B. Can insects help to ease the problem of world food shortage? Search 1975, 6, 261–262).  Note: Although Holt, cited by the authors, and Bequaert 1921, Bergier 1941 and Bodenheimer 1951 all pointed out that some insects were edible and were consumed by people, it was the 1975 paper, which for the first time linked global food shortages with direct or indirect insect consumption and it is for this reason that this pivotal work needs to be cited.

Incidentally, muslims regard locusts as edible (not only Jews do, which consider only  four species as edible).  Food taboos are mentioned on Line 332 and I recommend the authors find the Food Taboo review I got this info from in the J. Ethnobiol and Ethnomed.

Lines 334-361:The sagopalm grub (see the paper mentioned above that covers Papua New Guinea) is semi-domesticated by the Onabasulu in Papua New Guinea and wasps/hornets are not only seen as superior to most edible species of insect, but in parts of North-East India are also being domesticated (see: Kiewhuo, P.  Traditional rearing techniques of the edible Asian giant hornet (Vespa mandarinia Smith) and its socio-economic perspective in Nagaland.  Journal of Insects as Food and Feed, 2021).

Lines 363 etc.:  One must not go overboard with claims of nutritionally superior insects as food. In a thorough and reliable analysis Charlotte Payne et al. 2016 (Payne, C.L.R.; Scarborough, P.; Rayner, P.; Nonaka, K. Are edible insects more or less ‘healthy’ than commonly consumed insects? A comparison using two nutrient profiling models developed to combat over- and under-nutrition. Eur. J. Clin. Nutr.2016, 70, 285–291, doi:10.1038/ejcn.2015.149) have shown that insects are comparable in nutritional value to conventional food. The advantages of insets as a source of nutrients (protein, fatty acids, minerals, vitamins and even carbohydrates  (contrary to earlier belief Paoletti et al. 2007 have shown that 30% of humans CAN digest chitin, which is 50% carbs: Paoletti M.G.; Norberto,L., Damini, R. 2007. Human gastric juice contains chitinase that can degrade chitin. Ann. Nutr. Metab. 51, 244-251) is related to their environmental impact and feed conversion factor.

The world’s big problem is nowadays no longer a food shortage, but an overproduction of food, obesity and debility due to overeating. There are data available on the web to show the staggering amount of food wasted, thrown away, unused. Maybe the authors in one or two sentences could mention that, because it would make their review more balanced, more ’objective’. Please note:  About 17% of global food production may go wasted, according to the UN Environment Programme’s (UNEP) Food Waste Index Report 2021, with 61% of this waste coming from households, 26% from food service and 13% from retail and 8-10% of global greenhouse gas emissions are associated with food that is not consumed. This should not simply be ignored.

Author Response

This is a very well written, thorough and highly interesting and impressive manuscript. The authors cover historical facts that other reviews, with few exceptions, have in the past largely ignored.

We wish to thank the reviewer for appreciating our work. The reviewer has fully understood the reasons behind why we prepared this review, i.e., to show that consumption of insects by humans (now prospected as “novel food”) has its roots in the past also in the so-called Western societies, in the framework of other cultures.

I have rather few suggestions but ask the authors to consider them as acting on they would render their paper more balanced and complete. I

We thank the reviewer for the effort and time spent in giving us advice to improve the review, and we give our thanks also for providing clear suggestions. We did our best to address them.

N.B. The lines reported in our replies refer to the new version of the manuscript.

 I cannot fully understand why there is a need for a ‘Simple Summary” and an “Abstract”. That is most unusual and I suggest to keep either the Summary or the Abstract.

In the Guidelines, the journal recommends to include both a “Simple Summary” and an Abstract”

On Line 10 the authors forgot to include Papua New Guinea, Australia, and New Zealand! In fact, West Australian Aborigines thousands of years already began to semi-cultivate edible grubs by deliberately placing egg-laying female cerambycid beetles on suitable trees from which people could later harvest the edible insects (Reim, H. 1962. Die Insektennahrung der australischen Ureinwohner. Akademie Verlag, Berlin  -  cited in: Meyer-Rochow, V.B. and Changkija, S. 1997. Uses of insects as human food in Papua New Guinea, Australia, and North-East India: cross-cultural considerations and cautious conclusions. Ecology of Food and Nutrition 36,159-185).

We added “Oceania” both in the Simple Summary and Abstract, and we included mentions and references related to Oceanian countries (Aborigene culture) and also North East India in the text (lines 405-423, Ref. 5, 81,82,83,84). It is true that this important part of the entomophagy roots was missing from the review, and we thank the reviewer for making it notice and for giving us advice to fill the gap. However, the focus of our work are mainly the Western societies, but discussed in the wider context of the other cultures’ situation. To make the goal of our review clearer, we slightly modified the title

Line 55: the reference to ‘cows’ is a little odd, because to a Hindu, especially Brahmin, ALL animals and not just ‘cows’ are taboo and will not be eaten (I am married to an Indian lady!). There are reviews on Food Taboos in the Journal of Ethnobiology and Ethnomedicine, which could be of interest to the authors.

We added a sentence (lines 56-58) and a relevant citation  (#5)

On line 70 after “…..should also be considered” continue the sentence and add “…as Meyer-Rochow nearly 50 years had already suggested to FAO and WHO”.  (Meyer-Rochow, V.B. Can insects help to ease the problem of world food shortage? Search 1975, 6, 261–262).  Note: Although Holt, cited by the authors, and Bequaert 1921, Bergier 1941 and Bodenheimer 1951 all pointed out that some insects were edible and were consumed by people, it was the 1975 paper, which for the first time linked global food shortages with direct or indirect insect consumption and it is for this reason that this pivotal work needs to be cited.

Done. Thanks for the suggestion. Paper added (#8)

Incidentally, muslims regard locusts as edible (not only Jews do, which consider only  four species as edible).  Food taboos are mentioned on Line 332 and I recommend the authors find the Food Taboo review I got this info from in the J. Ethnobiol and Ethnomed.

We referred to Jews tradition (mentioned as an example, not meant to be comprehensive), because the Bible is  is part of the roots of the European culture. We added the Food Taboo Review (#5) to references (we cite it around the new version of the manuscript). The fact that only four species of locusts are regarded  as edible in the Jews tradition is controversial, as we understood also by the Review itself, so we did not mention this statement, but the reader may find it in the now cited Review.

We would like to emphasise that we explicitly refer to food taboos twice (lines 510-598) the paper (actually, insects are still a food taboo in the western culture), but this was not the main topic of our paper. We were particularly aware not to go deep in food taboo topics, especially when based on religious issues, because this goes beyond our paper’s goal. However, reference #5 may give much information to the readers interested to know more about this issue)

Lines 334-361:The sagopalm grub (see the paper mentioned above that covers Papua New Guinea) is semi-domesticated by the Onabasulu in Papua New Guinea and wasps/hornets are not only seen as superior to most edible species of insect, but in parts of North-East India are also being domesticated (see: Kiewhuo, P.  Traditional rearing techniques of the edible Asian giant hornet (Vespa mandarinia Smith) and its socio-economic perspective in Nagaland.  Journal of Insects as Food and Feed, 2021).

We have now specified at the beginning section 4 that our review focuses primarily on the Western societies. It was not meant to be comprehensive, especially as regards non-Western societies (which include very many examples of use of insects as human food) But it is certainly true that cross-cultural considerations are important when discussing the use of insects as human food, and Oceania and North East India must be included among the world areas where entomophagy has a very long tradition. We thank the reviewer to make us notice the gap. We included Oceania among the non-Western countries with traditions of insects as human food with relevant references and we also included the mentions to Papua New Guinea and North East India with relevant references (as mentioned in a previous reply, lines 405-423, ref. 5, 81-84.

Lines 363 etc.:  One must not go overboard with claims of nutritionally superior insects as food. In a thorough and reliable analysis Charlotte Payne et al. 2016 (Payne, C.L.R.; Scarborough, P.; Rayner, P.; Nonaka, K. Are edible insects more or less ‘healthy’ than commonly consumed insects? A comparison using two nutrient profiling models developed to combat over- and under-nutrition. Eur. J. Clin. Nutr.2016, 70, 285–291, doi:10.1038/ejcn.2015.149) have shown that insects are comparable in nutritional value to conventional food. The advantages of insets as a source of nutrients (protein, fatty acids, minerals, vitamins and even carbohydrates  (contrary to earlier belief Paoletti et al. 2007 have shown that 30% of humans CAN digest chitin, which is 50% carbs: Paoletti M.G.; Norberto,L., Damini, R. 2007. Human gastric juice contains chitinase that can degrade chitin. Ann. Nutr. Metab. 51, 244-251) is related to their environmental impact and feed conversion factor.

We added these contents and relevant references (lines 468-69, ref. 89, 90)

The world’s big problem is nowadays no longer a food shortage, but an overproduction of food, obesity and debility due to overeating. There are data available on the web to show the staggering amount of food wasted, thrown away, unused. Maybe the authors in one or two sentences could mention that, because it would make their review more balanced, more ’objective’. Please note:  About 17% of global food production may go wasted, according to the UN Environment Programme’s (UNEP) Food Waste Index Report 2021, with 61% of this waste coming from households, 26% from food service and 13% from retail and 8-10% of global greenhouse gas emissions are associated with food that is not consumed. This should not simply be ignored.

With respect for the reviewer, we do not totally agree with him/her. We rather see a big contradiction; on one side overproduction of food, obesity, debility due to overeating and food waste (with environmental consequences), but on the other side there are still people starving and dying because of lack of food.

But the reviewer is right: this issue has not to be ignored. We added two sentences (lines 633-639) (with references) in the Future Directions

We would like to emphasize that we did not wish to present the insects as a means to face food shortage, but, rather, we tried to stress their importance a sustainable protein (and other nutrient) source.

Reviewer 2 Report

 “Edible insects: a historical and cultural perspective on entomophagy”

This manuscript, under consideration for “Insects”, presents a survey on different aspects of edible insects or entomophagy. The focus is on different historical and cultural aspects of the use of edible insects, and their documentation in the literature. The topic is certainly interesting with an increasing relevance of edible insects as food and feed, relevant to food production, recourse utilisation, and climate impact. As such, it complements other reviews and reports on edible insects concerning the nutritional and legislative aspects or conditions of raising insects. Such an article focussed on the cultural and historical dimensions of the edible insects could be relevant for both researchers in the field and readers with a general interest in edible insects.

General comments:

My main concern with the manuscript in the present form is that it addresses very complex topics by historical and cultural dimensions, but does not present many facts on the evidence before written records, and lacks analysis relating to cultural anthropology or sociology, for example regarding the origins and tradigenesis of food taboos.

As an example, archaeological evidence is mentioned (lines 9, 511) but not discussed for any such details in the main text - what insects, what does this evidence reveal (occasional consumption vs regular consumption?), which locations, which era? Such discussion would be a feasible topic here. Also, there are several interesting details in the manuscript, but they are often a collection of single facts – see pg. 9, where single, or few, references are given at the end of single sections without depth on the available literature or integration and further critical discussion of references.

What is certainly lacking is a larger conceptual analysis. Authors should especially bring in a stronger emphasis on concepts relating to the cultural backgrounds, if they present a “cultural perspective”. How are insects ascribed a specific status or function, as a delicacy, or as currency? (latter: see line 517, again, not explained at all in the text). The discussion of taboo also has only very vague concept of either “the cultural taboo” (l. 447, as if a single doctrine) or “cultural taboos... against insects as a food source” (l. 534). This whole topic is rather underdeveloped, what defines a taboo in academic discourse and in this context, and importantly this specific taboo could/ should be formulated (and then referenced) in some way from a relevant resource.

See for another example of a discussion which is too general, lines 330-332:

Paul Rozin, an American psychologist born in 1936, wrote extensively about food and culture, and, in several of his works, he discussed the history and cultural significance of insect consumption [55].

This is a work many researchers in nutritional science, biology or entomology will likely not know about. In the present form, the section basically appears as name-dropping. What were these works, as you mention “several”, what is the finding of this discussion, in relation to the cultural perspective? We are left completely uninformed about the real analytic outcome of this work.

As a medical use is repeatedly mentioned, could this be backed up by some citations (e.g., the more detailed section on Aldrovandi), and are there references about the evaluation of such medical use from recent writings on history of medicine?

The manuscript has a great chance to present a concise and fascinating take on entomophagy beyond the current dominance of economic and ecological interests. In this sense, the final section on “Future directions” has considerable overlap with other recent reviews on the topic and somewhat distracts from the main focus of the sections before. In sum, this approach is an interesting contribution to the field of edible insects, but in its current form, it is underdeveloped and lacks a rigorous presentation of the relevant concepts.

Minor:

L 31, 32          here, the cultural perspective should briefly be defined.

L 41-47           introductory paragraph could already address insects specifically.

L 54 – 55        these are also examples why the text feels like an addition of material, withvery short mentioning. Can you explain how these special status came about, relating to insect cases?

L 72     entomophagy is a term with some facets, consider to include here Evans et al., ‘Entomophagy’: an evolving terminology in need of review, J Insects Food Feed 1(4): 293-305 (2015),

For historical development, see also Svanberg and Berggren, Insects as past and future food in entomophagic Europe, Food, Culture & Society 24: 624-638 (2021)

Several expressions are rather general, with no specific information on time or significance in the population, see e.g., l 449, 451, 468 (“some”) – please elaborate some more.

L 82/83           The paleontological dimension here could be developed with more detail.

L 94-96           This is repetitive to the section before.

L 119              …traditions [15].

L 130              remove first dot

L 187              displays two lateral horns on the thorax…

L 505, 542      both sections are numbered with “6”

L 521 – 525    this section seems not to fit into a “Conclusion”, rather present it earlier in the text.

L 569 – 573    This final section has only few references, consider to include other reviews on this topic like Ordonez-Araque et al., Edible insects for humans and animals: Nutritional composition and an option for mitigating environmental damage, Insects 13: 944 (2022)

L 694   Jongema, Y.

            Is this list still curated and updated, as it states the date from 2017?

Author Response

This manuscript, under consideration for “Insects”, presents a survey on different aspects of edible insects or entomophagy. The focus is on different historical and cultural aspects of the use of edible insects, and their documentation in the literature. The topic is certainly interesting with an increasing relevance of edible insects as food and feed, relevant to food production, recourse utilisation, and climate impact. As such, it complements other reviews and reports on edible insects concerning the nutritional and legislative aspects or conditions of raising insects. Such an article focussed on the cultural and historical dimensions of the edible insects could be relevant for both researchers in the field and readers with a general interest in edible insects.

We thank the reviewer for the overall appreciation of the paper topics. We made the best of our efforts to improve the review following his/her suggestions.

General comments:

My main concern with the manuscript in the present form is that it addresses very complex topics by historical and cultural dimensions, but does not present many facts on the evidence before written records, and lacks analysis relating to cultural anthropology or sociology, for example regarding the origins and tradigenesis of food taboos.

The topic of insect consumption by humans, its cultural and historical background and perspectives may be afforded by different points of views. In this manuscript submitted to “Insects”, it was afforded from the point of view of an entomologist (MLD) and  of a scholar of literature with a special (but not exclusive) interest in ancient Greek and Roman literature, history and language(MO). The topic is fascinating and very wide, hard  to deal with in all its aspects in a review. We, however, tried to be as comprehensive as possible, even more in the revised version of the MS , also following the reviewer’s suggestion.

As regards evidence of human entomophagy before written records, at the beginning of section n. 2 we added a whole paragraph with relevant references (lines 88-115), mainly dealing with cave paintings, with mentions to coprolites referred to the European area (aur main target) and (for coprolites), the Mexican area.

We added a reference (#5) about food taboos involving insects. The analysis on cultural anthropology or sociology, regarding the origins and tradigenesis of food taboos is a fascinating topic, certainly not involving only insects and, with respect for the reviewer, we find it a little beyond the purpose of our review, which is focalized on insects, on the fact that insect consumption by humans has its roots in the past even in the Western  society, with particular  (but not exclusive) reference to entomophagy in ancient Greece and Rome. Insects are presented as “novel” food, bur they may be regarded also as part of our cultural tradition.

As an example, archaeological evidence is mentioned (lines 9, 511) but not discussed for any such details in the main text - what insects, what does this evidence reveal (occasional consumption vs regular consumption?), which locations, which era? Such discussion would be a feasible topic here. Also, there are several interesting details in the manuscript, but they are often a collection of single facts – see pg. 9, where single, or few, references are given at the end of single sections without depth on the available literature or integration and further critical discussion of references.

As said above, we added some archaeological evidence (cave paintry, coprolites) revealing that insects were part of the diet of prehistoric populations, with locations and era (see the beginning of section 2, lines 88-115). We could not find evidence of a regular vs. occasional consumption, something that archaeologists may possibly discuss in a different context. Archaeological evidence of insect consumption during Ancient Greek and Roman times (the historical period we mainly addressed) is (to the best of our knowledge) lacking, contrary to written records  and may not contribute to provide a comprehensive understanding of the extent or cultural significance of entomophagy.

We tried to include the references (14-23) that could support our statements and provide the reader with further sources, if he/she wishes to go deeper into a topic.

With respect for the reviewer , in some cases, in our opinion, 1-2 references per statement  may be sufficient, and too many may confuse the reader. But we added other references, where the reviewer(s) made specific suggestions. The references were 90 in the former version and are now 120.

What is certainly lacking is a larger conceptual analysis. Authors should especially bring in a stronger emphasis on concepts relating to the cultural backgrounds, if they present a “cultural perspective”. How are insects ascribed a specific status or function, as a delicacy, or as currency? (latter: see line 517, again, not explained at all in the text). The discussion of taboo also has only very vague concept of either “the cultural taboo” (l. 447, as if a single doctrine) or “cultural taboos... against insects as a food source” (l. 534). This whole topic is rather underdeveloped, what defines a taboo in academic discourse and in this context, and importantly this specific taboo could/ should be formulated (and then referenced) in some way from a relevant resource.

We did not expect to cover the whole, very wide topic of the roots of entomophagy and its perspectives. This (fascinating) goal goes beyond the purpose of our review, which is aimed at showing how entomophagy has its roots in the past also in the Western societies, as also proven by the references found in the Greek and ancient Roman literature (an aspect that, to the best of our knowledge, has been poorly afforded before). We also made it notice how, in the Western culture, interest in entomophagy decreased (but never disappeared) after the fall of the Roman Empire, during Middle Age, Renaissance and in the modern age. Now insects, in Western societies, are presented as “novel” food (with good perspectives), but they have indeed been part of our cultural and culinary tradition for a long time now (even though not at the same level as in  some non-Western societies). We have made our best efforts to develop the above- mentioned topics. We have tried to address all the reviewer’s remarks to improve our review following the suggestions, we added several references, and we hope to have improved the presentation of the concepts.

Regarding specific remarks: we deleted the reference to insect as “currency” (line 517 in the former version)

About taboos, we modified the sentence in lines 598 and following as: “Despite cultural taboos (rooted in cultural, religious, or personal beliefs) and societal biases against insects as a food source [5], many cultures worldwide have traditionally consumed insects for their nutritional and medicinal properties “. Reference n. 5 (also cited in the introduction of the new version) is  “Meyer-Rochow, V. Food Taboos: Their Origins and Purposes. Journal of ethno. and ethnomed. 2009, 5, 18, doi:10.1186/1746-4269-5-18 “

See for another example of a discussion which is too general, lines 330-332:

“Paul Rozin, an American psychologist born in 1936, wrote extensively about food and culture, and, in several of his works, he discussed the history and cultural significance of insect consumption [55].”

This is a work many researchers in nutritional science, biology or entomology will likely not know about. In the present form, the section basically appears as name-dropping. What were these works, as you mention “several”, what is the finding of this discussion, in relation to the cultural perspective? We are left completely uninformed about the real analytic outcome of this work.

We deleted the citation of Paul Rozin, not really necessary.

Our review is focused on emphasizing the interest of entomophagy in the western world where it has roots in the past and in the history, and citing scholars from different time periods who have discussed insect consumption can be a powerful way to convey the cultural perspective and highlight the historical and cross-cultural significance of this practice. Historical contexts, in fact, demonstrate that insect consumption is not a recent phenomenon, and it shows that this practice has deep roots and has been part of human culture and cuisine for centuries. We added a couple of sentences about this topic at the end of section 3 (lines 368-372).

As a medical use is repeatedly mentioned, could this be backed up by some citations (e.g., the more detailed section on Aldrovandi), and are there references about the evaluation of such medical use from recent writings on history of medicine?

As regards the medical use of insects we already reported several references.

In detail, about the medical use of insects in ancient Rome we cited:

  • Wallace-Hadrill, A. Pliny the Elder and Man’s Unnatural History. Greece & Rome 1990, 37, 80–96. [34]
  • Plinius Secundus, G. Natural History. 1: Preface and Books 1-2 / with an Engl. Transl. by H. Rackham; The Loeb classical library; [Repr. der Ausg.] 1938, repr. with corr. 1944, rev.repr. 1949.; Harvard Univ. Press: Cambridge, Mass., 2007; pp. 1-400 (ISBN 978-0-674-99364-8). [35]
  • Aelian; McNamee, G. {Aelian’s} on the Nature of Animals; Trinity University Press: San Antonio, Tex., 2011; pp. 1-180 (ISBN 978-1-59534-111-2).[43]

About the medical use of insects in the Reinassance period we reported:

  • Gudger, E.W. The Five Great Naturalists of the Sixteenth Century: Belon, Rondelet, Salviani, Gesner and Aldrovandi: A Chapter in the History of Ichthyology. Isis193422, 21–40. [62]

About the medical use of insects in Africa we cited:

  • Hlongwane, Z.T.; Slotow, R.; Munyai, T.C. Indigenous Knowledge about Consumption of Edible Insects in South Africa. Insects202112, 22. [72]
  • Van Huis, A.; Van Itterbeeck, J.; Klunder, H.; Mertens, E.; Halloran, A.; Muir, G.; Vantomme, P. Edible Insects. Future Prospects for Food and Feed Security; FAO: Rome, Italy, 2013; Volume 171, ISBN 978-92-5-107595-1.[73]

In the new version in lines 401, We added:

Cartay, R.; Dimitrov, V., Feldman, M. An Insect Bad for Agriculture but Good for Human Consumption: The Case of Rhynchophorus palmarum: A Social Science Perspective. 2020 DOI: 10.5772/intechopen.87165 [76]

And in line 600  we added a review:

Meyer-Rochow, V.B. (2017). Therapeutic arthropods and other, largely terrestrial, folk-medicinally important invertebrates: a comparative survey and review. J. Ethnobiol  13:9 (31 pages) https://doi.org/10.1186/s13002-017-0136-0 [114]

The manuscript has a great chance to present a concise and fascinating take on entomophagy beyond the current dominance of economic and ecological interests. In this sense, the final section on “Future directions” has considerable overlap with other recent reviews on the topic and somewhat distracts from the main focus of the sections before. In sum, this approach is an interesting contribution to the field of edible insects, but in its current form, it is underdeveloped and lacks a rigorous presentation of the relevant concepts.

We take note that the reviewer appreciated the review main topic. We would like to further emphasise that we did not expect to cover the whole (please see our reply to our first general comment) Again, we have tried to address all the reviewer’s remarks to improve our review following his/her suggestions, we added several references, and we hope to have improved the presentation of the concepts. The “Future directions” may have overlaps with other recent reviews (one of which has been added to the references, following the reviewer’s suggestions) but, still, it discusses perspectives that, in our opinion, are worth to be outlined also here (including the possibility that insects become a mainstream food, even as derivatives/ingredients of different food products, the fact that they are a sustainable source of animal proteins). Yet this “novel” food has its roots in the past also in the Western societies, as highlighted in the final sentence, as one of the “take home” messages of our review.

Minor (the line number has changed in the revised MS)

L 31, 32          here, the cultural perspective should briefly be defined.

Done

L 41-47           introductory paragraph could already address insects specifically.

With respect for the reviewer, we would like to address insects in a wider context, i.e., addressing briefly other animals.

L 54 – 55        these are also examples why the text feels like an addition of material, withvery short mentioning. Can you explain how these special status came about, relating to insect cases?

We already mentioned that in ancient Egypt the scarab beetle (associated with the god Khepri) represented rebirth and renewal. Following the suggestions of another reviewer, we specified other animal food cases non permitted in Hinduism, especially for Brahmins. These are not examples relating to insect cases (which we did not find), yet we believe that some examples concerning non-insect animals may be worth, in a wider context

L 72     entomophagy is a term with some facets, consider to include here Evans et al., ‘Entomophagy’: an evolving terminology in need of review, J Insects Food Feed 1(4): 293-305 (2015),

The reviewer is right. We added a few words explanation and the reference

For historical development, see also Svanberg and Berggren, Insects as past and future food in entomophagic Europe, Food, Culture & Society 24: 624-638 (2021)

Added at the end of the Introduction

Several expressions are rather general, with no specific information on time or significance in the population, see e.g., l 449, 451, 468 (“some”) – please elaborate some more.

Line 513: added two examples of EU countries which have allowed the sale of certain insects. We need to be cautious, because the situation in EU is not well defined

Line 515: “some traditional dishes” are elaborated a few lines below, e.g. different types of chees colonized by insect larvae

Line 532: “and is considered a delicacy by some people”. In our opinion, it is not worth to elaborate more: “some” people consider a delicacy Casu Marzu, but we cannot be more precise, since this cheese can be consumed by producers, but cannot be sold. The situation is actually confused.

L 82/83           The paleontological dimension here could be developed with more detail.

The paleontological dimension has been widened, lines 88-115 (with references) in the new version

L 94-96           This is repetitive to the section before.

Sentence and reference deleted and merged in the paragraph before

L 119              …traditions [15].

Done. Thanks.

L 130              remove first dot

Done. Thanks

L 187              displays two lateral horns on the thorax…

Accepted

L 505, 542      both sections are numbered with “6”

Amended. Thanks.

L 521 – 525    this section seems not to fit into a “Conclusion”, rather present it earlier in the text.

We deleted the sentence, which presented concepts already expressed in section 2

L 569 – 573    This final section has only few references, consider to include other reviews on this topic like Ordonez-Araque et al., Edible insects for humans and animals: Nutritional composition and an option for mitigating environmental damage, Insects 13: 944 (2022)

Done

L 694   Jongema, Y.

            Is this list still curated and updated, as it states the date from 2017?

To the best of our knowledge, this is the more recent list of edible insects. We changed the link, because the one that was indicated in the previous version of the paper was seemingly not working

Reviewer 3 Report

I find this work interesting and worth publishing, especially since it prefaces interesting historical information about the relationship between humans and insects in various cultures in past times. The authors have done a very good and useful job quoting historical sources. It is an excellent tool for combating entomophagy, especially in the context of European societies. I didn't find any big errors.

Author Response

We thank the reviewer for the nice appreciation shown for our work, of which he/she has well understood the goal (i.e. to show how edible insects now presented as “novel” food and prospected as a “future” food have their roots as such in the past (with major reference to the Greek-Roman culture) also in the Western societies (in particular in Europe)

Round 2

Reviewer 1 Report

Congratulations!

You have prepared a well-researched and balanced review.

Minor editorial attention to the English would be desirable.

Author Response

Comments and Suggestions for Authors

Congratulations!

You have prepared a well-researched and balanced review.

We thank the reviewer for the valuable comments which allowed us to improve the review

Comments on the Quality of English Language

Minor editorial attention to the English would be desirable.

The review has been carefully re-read to check English and the Journal itself will edit the English of the final version. 

Reviewer 2 Report

The revised version now incorporates additional material, and the response letter details the changes incorporated in the review. Overall, I think this is a fascinating topic, and the historical material is well collated, while my main approach was that a “cultural perspective” raises the expectation of an analysis from a cultural angle and with the respective methodological considerations. In this sense, the response that “its cultural and historical background and perspectives may be afforded by different points of views“ does not make it more obvious which cultural aspects are analysed, and why/ how. This, in my perception, is not a topic of narrowing down „the whole, very wide topic of the roots of entomophagy and its perspectives”. If this review added some more analysis of the cultural mechanisms (again, take the example of the origin and tradigenesis of a food taboo), this would strengthen the lasting relevance of this work, since they are not regularly considered by biologists. The new material on the archaeological findings are a good example of such interesting material. Yet, such cultural concepts are not defined here (taboo is not defined or explained, to keep with this example). It is the choice of the editors if the current form of the analysis is adequate for the journal. 

In addition, please consider some specific points:

The cultural perspective, again, should be made explicit – for reasons relating to ecological (conservation), nutritional (health education), religious (spiritual), commercial (economic) or trending (socio-psychological) reasons etc. (l. 32) – this would be something for the main text to discuss.

How do the Hindu food taboos link to insects? (l. 55)

“Shudraof” – should be “Shudra”? (l. 57)

Tettigonia is not a caverniculous insect genus? (l. 104)

Could you include some figures on the cave paintings/ carvings? Or any further artwork, e.g., from the cited sources?

How reliable are these accounts on four-legged insects as historical entomological “knowledge”? (l. 126)

The work by Paul Rozin did sound highly relevant, and should be covered here, but what are the relevant findings? Has this work been taken up in this historical - cultural context?

Author Response

The revised version now incorporates additional material, and the response letter details the changes incorporated in the review. Overall, I think this is a fascinating topic, and the historical material is well collated, while my main approach was that a “cultural perspective” raises the expectation of an analysis from a cultural angle and with the respective methodological considerations. In this sense, the response that “its cultural and historical background and perspectives may be afforded by different points of views“ does not make it more obvious which cultural aspects are analysed, and why/ how. This, in my perception, is not a topic of narrowing down „the whole, very wide topic of the roots of entomophagy and its perspectives”. If this review added some more analysis of the cultural mechanisms (again, take the example of the origin and tradigenesis of a food taboo), this would strengthen the lasting relevance of this work, since they are not regularly considered by biologists. The new material on the archaeological findings are a good example of such interesting material. Yet, such cultural concepts are not defined here (taboo is not defined or explained, to keep with this example). It is the choice of the editors if the current form of the analysis is adequate for the journal.

The cultural perspectives have been now been added in lines 31 and following, as suggested by the reviewer, and as also highlighted below (see the reply to the first of the specific comments)

With respect for the reviewer, we believe that, for a review on entomophagy focused on Western societies, a detailed analysis of the origin and tradigenesis of a food taboo (which do not concern insects only), would be excessive, and beyond the journal’s goals (though the topic is certainly fascinating). This tradigenesis is, however, well afforded in reference n. 5 (Meyer-Rochow, V. Food Taboos: Their Origins and Purposes. Journal of ethno. and ethnomed. 2009, 5, 18, doi:10.1186/1746-4269-5-18).

In addition, please consider some specific points:

The cultural perspective, again, should be made explicit – for reasons relating to ecological (conservation), nutritional (health education), religious (spiritual), commercial (economic) or trending (socio-psychological) reasons etc. (l. 32) – this would be something for the main text to discuss.

We further modified the sentence (lines 31 and following) as: “These topics are afforded in this article from a historical and cultural perspective (e.g., ecological, nutritional, spiritual and socio-psychological), with a focus on the progressive acceptance of edible insects in Western societies. since this novel food has also its roots in the Western world.”

Among the cultural perspectives suggested by the reviewer we chose those more in line with the review content and aims.

How do the Hindu food taboos link to insects? (l. 55)

We gave a few examples of food taboos unrelated to insects to contextualize the topic in a broader scope, also to follow the advice of another reviewer.

“Shudraof” – should be “Shudra”? (l. 57)

Yes, we changed. Thanks

Tettigonia is not a caverniculous insect genus? (l. 104)

The genus of the “cave grasshopper” is widely uncertain. We deleted Tettigonia sp.. Thanks.

Could you include some figures on the cave paintings/ carvings? Or any further artwork, e.g., from the cited sources?

Unfortunately, we don’t hold original pictures of figures on the cave paintings. The readers may, however, find such figures directly in the cited sources.

How reliable are these accounts on four-legged insects as historical entomological “knowledge”? (l. 126)

These accounts on four-legged insects are not reliable as historical entomological “knowledge”.

In lines 138 and following we provide some hypotheses about the reasons why insects “with four legs” are mentioned in the Bible.

The work by Paul Rozin did sound highly relevant, and should be covered here, but what are the relevant findings? Has this work been taken up in this historical - cultural context?

We re-inserted Pail Rozin’s an cooperator work and briefly discussed some of the implications of his work in a cross-cultural perspective, with a relevant reference